# On Sequence Segmentation with overlapped Chunks in Machine Learning

## Abstract

Operating on very long sequences can be problematic for many sequence modelling methods like Transformers or recurrent neural networks. To avoid this issue, long sequences are often split into smaller chunks instead. For various reasons, these chunks typically are overlapped with each other which causes an increase in tensor size by however much the chunks are overlapping.

This paper attempts to find a better understanding on overlapped sequence chunks and what they accomplish. Specifically, the focus of this paper is on audio inputs in both the time and frequency domain. Previous models for speech separation and audio super resolution which use overlapped chunks are modified to allow for reduced or even removed overlaps which causes significant decreases in computational cost while maintaining accuracy.

## 1    Introduction

Certain types of data, like audio, consist of very long sequences. For these data types, it is not viable to perform sequence modelling techniques such as Transformers (Vaswani et al., 2017). It is necessary and common practice to apply these sequence modelling steps to much shorter sequences instead.

One way to accomplish this is through downsampling. This is especially popular for tasks such as automatic speech recognition (Baevski et al., 2020; Zeyer et al., 2021), audio classification (Kong et al., 2019) and many other problem areas where Transformers are used (Dosovitskiy et al., 2020). The wav2vec2 model (Baevski et al., 2020), for example, reduces the sequence length to less than 1% of its original sequence length before applying Transformers to it. However, such aggressive downsampling is not possible for all problem areas. The reason it works for speech recognition and audio classification is that their input size is orders of magnitude larger than their output size and therefore information can be summarized.

For other problem areas, mainly those which output not a class or text but audio itself, downsampling that aggressively is not possible. These problem areas include audio generation, speech synthesis, source separation, speech enhancement and audio super resolution. Downsampling to the same degree as the speech recognition and audio classification models is not viable since it removes too much information which will negatively impact the accuracy of the model.

Therefore, a different solution is necessary to use sequence modelling techniques like Transformers. The basic solution is to split the sequence into equal-sized chunks and then apply sequence modelling to the much shorter sequence of the chunks. This, however, causes a new issue. The context that exists between the chunks is lost which will negatively impact the accuracy of the model. To address this, the sequence is split into overlapping chunks instead. Overlapping chunks will ensure that no context at the chunk edges is lost. It does, however, also have a negative side effect: due to the overlap, the new tensor will contain duplicate information. For an overlap ratio of 50%, for example, the tensor size nearly doubles since every sequence element except for the beginning of the first chunk and end of the last chunk are contained twice in the new tensor. Figure 1 shows an example of time domain chunking with a 50% overlap. Note, that we use the terms window size and chunk size interchangeably. The window size or chunk size determines the number of samples per chunk while the hop size describes how far we move along the original sequence for each chunk. The smaller the hop size in comparison to the chunk size, the greater the overlap.

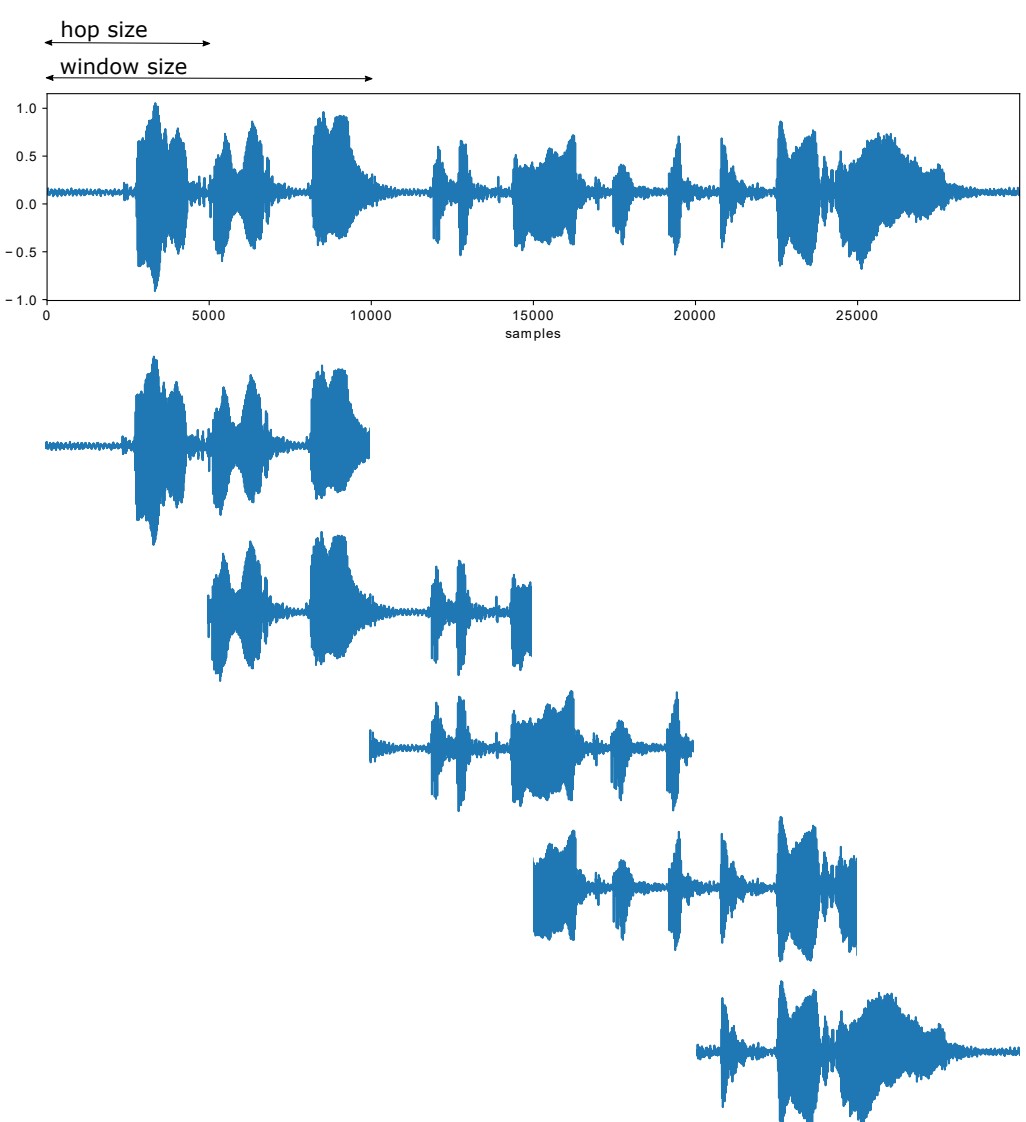

Figure 1: Splitting the sequence into chunks with a window size of 10000 samples and a hop size of 5000 samples, resulting in a 50% overlap. The total number of samples increases from 30000 to 50000 as every sample except the first half of the first chunk and the last half of the last chunk is represented twice.

Since this sequence segmentation step turns a one dimensional tensor into a two dimensional tensor, it is common for time domain models to apply sequence modelling techniques like Transformers or RNNs to both axes (Luo et al., 2020; Chen et al., 2020; Subakan et al., 2023; Jiang et al., 2024). To reverse the sequence segmentation after the sequence modelling has been performed, the overlapped bits are added to each other and the sequence in its original shape is reassembled.

While there are many models which operate on overlapped chunks in the time domain (Subakan et al., 2021; Luo et al., 2020; Chen et al., 2020), the most common signal processing operation which utilizes overlapping chunks is the short-time Fourier transform (STFT). Just like previously described, the signal in the time domain is split into overlapping chunks. However, this time, a window function is applied followed by a fast Fourier transform (FFT) for each chunk. Figure 2 shows each step of the STFT process.

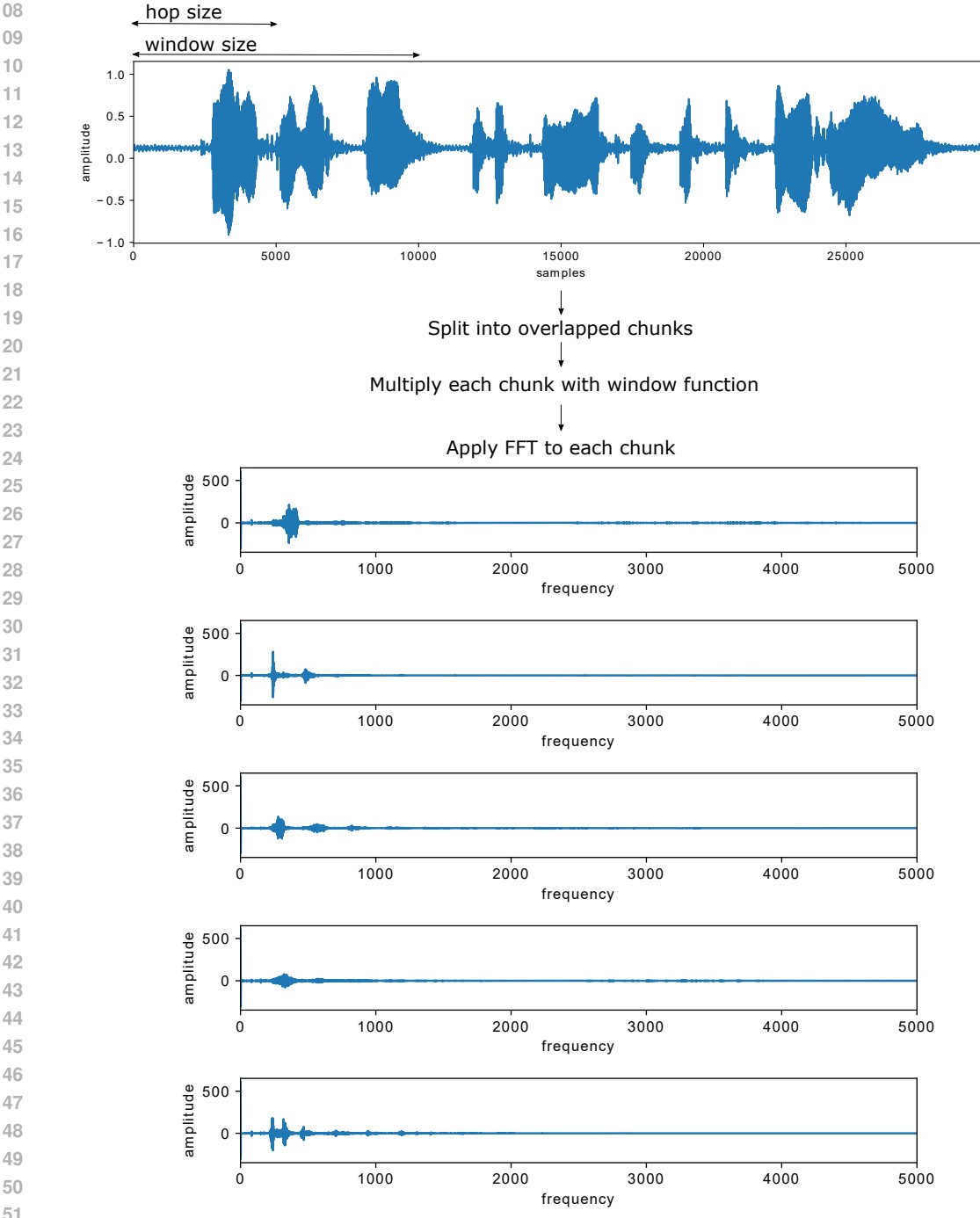

Figure 2: Overview of the short-time Fourier transform (STFT).

The reasoning behind choosing overlapping chunks for the STFT is different from the time domain chunking. The Fourier transform assumes that the signal is periodic. Real life signals, however, generally are not periodic. This causes a phenomenon called spectral leakage which in turn leads to distortion. To minimize spectral leakage, the signal is multiplied with a window function whose purpose it is to taper off the start and end of the signal to 0 making the signal somewhat periodic. Setting the beginning and end to 0, however, means that we lose said information - which is why

overlap is utilized for the STFT. Having duplicates of all the sequence samples that are set to 0 to make the chunks periodic ensures that no information is lost.

Applying the window function means that there is no duplicate information. And without using the window function, the STFT will not be able to be reversed perfectly. Therefore, removing overlaps in the frequency domain is more difficult than in the time domain. While entirely removing overlaps from STFT-based models may not always be possible or advisable, a reduction of overlaps still is. Almost all of the recent SOTA models which use the STFT use an overlap ratio of 75% or greater (Wang et al., 2022; Han & Lee, 2022; Luo & Yu, 2022; Bai et al., 2024; Abdulatif et al., 2024; Liu et al., 2022; Yuan et al., 2024; Xue et al., 2024; Liu et al., 2023; Zhao et al., 2021; Lu et al., 2024; Lee et al., 2024a; gil Lee et al., 2023; Lee et al., 2024b; Liao et al., 2024; Liu et al., 2024; Shibuya et al., 2024). The reason why this is a popular choice despite quadrupling the tensor size is that it maximizes the spectral and temporal resolution of the STFT. Choosing a large window size maximizes spectral resolution while choosing a small hop size maximizes temporal resolution.

This paper contains the following contributions:

1. Gaining a better understanding of overlapped chunks in the context of neural networks and what makes them effective.

2. Suggesting multiple strategies which can remove overlaps.

3. Testing said strategies on two models which operate in different domains and on different problem areas. Removing overlaps from these models results in major decreases to computational cost while maintaining accuracy.

## 2 REMOVING OVERLAP

In this section, we describe the necessary steps to remove overlaps without losing accuracy. While there are multiple ways to remove overlaps, they are all based on the same principle. The cause that necessitates overlaps is the fact that the sequence is split into chunks at the same spots. For time domain models, this means that context between chunks is lost and for frequency domain models, this means that the window function would set the same parts of the sequence to 0 and therefore information is lost. The way to fix the overlap issue is to split the sequence at different points. This assumes that the underlying machine learning model consists of multiple layers, meaning we can apply a fraction of the total layers to each variation of different sequence segmentation.

### 2.1 SHIFTING THE SEQUENCE

The basic approach to removing overlap is to shift the sequence before each sequence modelling layer, then split it up into chunks, apply the sequence modelling, reassemble the sequence and then undo the sequence shift. By choosing different sequence shifts each time, we get to keep the sequence context that would otherwise be lost for time domain models and we do not lose the information from the window function for the STFT.

Shifting the sequence solves the core issue since it no longer splits the sequence at the same points. The main consideration for the shifts is that they should not be a multiple of the chunk size. If they were, they would split the sequence at the same spots each time, defeating the purpose of this step.

The main advantage of this strategy is that it allows us to keep the chunk size consistent. The main disadvantage is that it causes issues for the first or last chunk depending on the shift direction since padding is not used as the objective of this operation is to reduce tensor size. Shifting the sequence will cause samples of the first and last chunk to mix with each other which is an unwanted side effect.

### 2.2 VARIABLE CHUNK SIZES

Another way of removing overlaps is to use different chunk sizes. This idea also assumes that the sequence segmentation has to be done and undo for each sequence modelling layer. By using different chunk sizes, especially ones that are not multiples of each other, the splits happen at different places in the original sequence.

As is the case with sequence shifting, the different chunk sizes should not be multiples of each other since some of their splits would be identical.

Using multiple chunk sizes has the advantage that it can capture additional patterns due to the changing of the window size and number of chunks for each sequence modelling layer. Compared to shifting the sequence, however, it has the disadvantage of potentially negatively impacting the computational cost by changing the sizes of the sequential axes where sequence modelling layers are applied to. Unlike shifting the sequence, however, this strategy does not cause non neighboring samples to end up in the same chunk.

### 2.3 INCREASING PARAMETERS

Shifting the sequence or using variable chunk sizes is the main step which allows for the removal of overlaps. However, if one uses an identical model, once with overlap and once without overlap with shifting or variable chunk sizes instead, the model using overlap will always achieve greater accuracy. This is because fundamentally, the model using overlap applies sequence modelling to its input more often than the model not using overlaps. Specifically, a 50% overlap doubles sequence modelling steps for the same model while a 75% overlap quadruples sequence modelling steps. Working on overlapped chunks is almost a form of parallelization.

While a 50% overlap does apply the sequence modelling twice per layer, these two applications have no awareness of each other. It is therefore no true parallelization, as the overlaps just get added to each other during the overlap and add step when the sequence segmentation is undone. Compared to a model which does not split the sequence up at all, it would be equivalent to stacking the sequence in the batch dimension at the beginning and then adding them up at the end. This type of model architecture is not used, however, because it is clearly inefficient. For the same computational budget, it makes more sense to increase other parts of the model, like the total amount of sequence modelling layers or the channel size.

Going from a 50% overlap to no overlap will roughly halve the computation time and memory usage and would therefore allow to double certain hyperparameters like the channel size or sequence modelling blocks without getting slower or using more memory. It is also possible to only slightly increase the hyperparameters of a model without overlap to match the original's accuracy while being significantly faster and more memory efficient.

To summarize, not using overlaps means that the model is using a fraction of the sequence modelling steps as a model using overlaps. Since these extra sequence modelling steps of the model with overlap have no awareness of each other, however, they are not optimal. Instead, having each sequence modelling step be applied in sequence is more effective.

## 3 EXPERIMENTS

In order to evaluate the no overlap approach, we implement it on existing architectures for two different tasks, single channel speech separation and audio super resolution. The architectures we tested are the SepFormer (Subakan et al., 2021) which is a time domain speech separation model and the NU-Wave2 (Han & Lee, 2022) which is a super resolution model which uses the STFT. We selected these models to include multiple problem areas and to show how our approach performs during both time and frequency domain models. Another reason we chose these models is that they are fairly recent and still close to SOTA performance while having public code which we used and adjusted for our experiments.

### 3.1 DATASETS

We test on the most common benchmarks for both speech separation and audio super resolution.

For speech separation, we use the WSJ0-2Mix (Hershey et al., 2016). This dataset is based on the WSJ0 corpus (Garofolo, John S. et al., 1993). The WSJ0-2Mix is a two speaker separation dataset without background noise and without reverberation. It contains 30 hours of training, 10 hours of validation and 5 hours of evaluation data. 119 different speakers with roughly half being female and the other half being male are included. Different utterances but the 101 same speakers are used for

the training and validation sets while the evaluation set has both different utterances and 18 different speakers than the training and validation sets.

The input of the WSJ0-2Mix task is a single channel mixture where two speakers talk over each other. These mixtures are artificially created by using the recordings of the individual speakers. The goal is to recover the original utterance for each speaker out of the mixture, meaning the outputs are two single channel waveforms which estimate the original utterances. We use the same setup of the data as the original SepFormer, meaning we use the 8 kHz version of the data and the same data augmentation methods.

For audio super resolution, we use the VCTK corpus (Yamagishi et al., 2019). It contains 44 hours of data from 108 different speakers. The first 100 speakers are used for training while the last 8 are used for evaluation. The inputs are downsampled to a fraction of its original sampling rate of 48 kHz and then upsampled back to 48 kHz. The goal of the super resolution model is to recover the lost information in the upper frequencies.

### 3.2 SPEECH SEPARATION - SEPFORMER

The SepFormer model (Subakan et al., 2021) is a speech separation model which operates in the time domain on overlapped chunks. The key part of the model which performs the sequence modelling and speech separation are 32 Transformers. Half of them are applied to the sequence of neighboring samples inside the chunks (intra-processing) and the other half are applied to the sequence of chunks (inter-processing). We show a simplified overview of the SepFormer architecture in Figure 3.

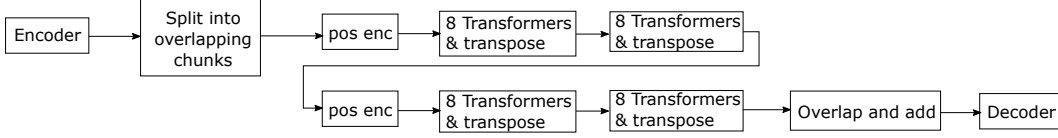

Figure 3: Overview of the SepFormer architecture.

In order for the SepFormer architecture to work without overlaps, we need to make some adjustments to its architecture. The overall adjusted SepFormer architecture is shown in Figure 4 and Figure 5. Note, that in the original SepFormer, sequence segmentation and overlap and add are performed only once while in the adjusted architecture this step is performed once per Transformer, meaning 48 times in total.

The first step to make the removal of overlaps work is to shift the sequence before each Transformer as was described in section 2.1. We increase the shifts by 10 for each Transformer starting from 0. In our experiments changing the shift value had only minor impact on accuracy as long as it was not too small in reference to the chunk size.

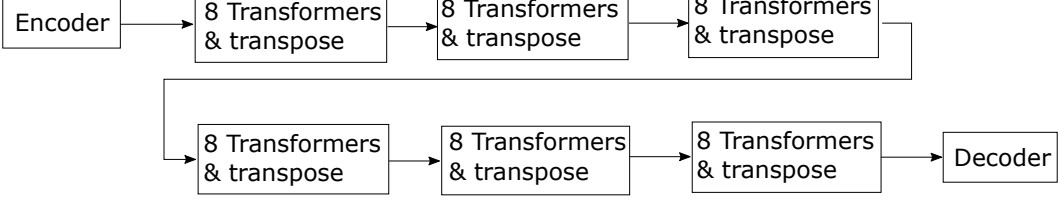

Figure 4: Overview of the adjusted SepFormer architecture without overlap.

Second, we need to change the way positional encoding is handled. In the original Transformer architecture, positional encoding is applied once per block of 16 Transformers across the axis of neighboring samples. Due to the shifting of the sequence which happens before each Transformer, we cannot apply positional encoding in the same way. The way positional encoding is handled

in the original architecture does not account for undoing and redoing the sequence segmentation which is necessary for using the sequence shifting. The reason we have to change the way positional encoding is integrated is because the sequence shifting causes samples to move across chunks which means that they change position, therefore defeating the purpose of positional encoding.

Applying positional encoding to the entire sequence is not a viable alternative either since the actual sequence modelling only operates on the much smaller chunks and sequence of chunks. Therefore, positional encoding needs to be applied for each Transformer instead, meaning it is applied 24 times more often than in the original architecture. Since positional encoding is not an expensive operation, however, this is not problematic in terms of computational cost. It does require another adjustment since adding 24 times more positional embeddings negatively impacts accuracy because they distort the output. In order to fix this issue, we simply subtract the positional embeddings at the end of each Transformer. We show this step in Figure 5 under the term positional decoding which is placed directly after the Transformer. The positional encoding is only necessary for the self-attention layer and can therefore be safely removed afterwards.

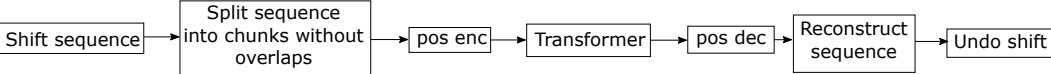

Figure 5: Architecture of the Transformers in the adjusted SepFormer model.

The third adjustment is to choose different chunk sizes for the intra- and inter-processing. This step is not technically necessary, but it allows for a closer comparison between our adjusted SepFormer architecture and the original model. In the original model, sequence segmentation only happens once before the 32 Transformers. The SepFormer model uses a chunk size of 250. If the full sequence contains 4000 samples, using a 50% overlap with a chunk size of 250, we get 31 total chunks. Therefore, for the original SepFormer, the sequence axis for the Transformers doing intra-processing has a size of 250 and the sequence axis for the Transformers doing inter-processing has a size of 31.

Since the adjusted version does not use overlaps, we only produce half the number of chunks, in the example from above it would be 16, for the same chunk size. This effectively halves the size of the sequence axis of the inter-processing Transformers. To get the same size of the sequence axis for the inter-processing Transformers, we need to halve the chunk size for the inter-processing Transformers for their sequence segmentation. This is not an issue, however, since we disassemble and reassemble the sequence for each Transformer anyway due to the shifts.

For our experiments we simply set the chunk sizes to 250 for the intra-processing Transformers and 125 for the inter-processing Transformers to match the sequence lengths of the original model which will allow for a better comparison. It is, however, also possible to change the chunk size for each Transformer instead which would make the shift sequence step unnecessary because if the sequence splits happen at different spots in the original sequence, then no context would be lost. While this would be the more practical solution, it also makes comparisons with the original model less meaningful since any change could be caused by the difference in the sizes of the sequence axes.

The fourth and final adjustment to the SepFormer architecture is to simply increase the total number of Transformers. This was already mentioned in section 2.3, but due to having a 50% overlap, each Transformer in the original SepFormer architecture performs sequence modelling twice on the original sequence. Since both of these applications are unaware of each other, however, they are not as effective as if they were just applied one after another.

This is the main contribution of our paper. Overlapped chunks are not optimal because it would always be preferable to apply sequence modelling steps in sequence since this way they are aware of each other and therefore much more effective. If our adjusted model had the same amount of Transformers it would have reduced accuracy compared to the original model because we only perform half the sequence modelling steps since in the adjusted model each Transformer performs sequence modelling only once on the original sequence. If we increase the number of Transformers by 50%, we perform 75% of the sequence modelling steps compared to the original architecture.

Our adjusted model contains 48 total Transformers as is shown in Figure 4 while the original model only contains 32 total Transformers.

We show the results of our experiments on the WSJ0-2Mix comparing the original SepFormer with our adjusted version in Table 1. Speech separation accuracy is measured in scale-invariant signal-to-distortion ratio improvement (SI-SDRi) Roux et al. (2018) where greater values represent better accuracy.

Table 1: Comparing the model size, scale-invariant signal-to-distortion ratio improvement (SI-SDRi), computation time and memory usage on the WSJ0-2Mix for models using overlap and models not using overlap. Computation time and memory usage were measured for inputs with a length of 32000. The computation time refers to how long the models take to operate on 1000 inputs.

| Method | Overlap | Model size | Computation time (s) | | Memory usage (GB) | | SI-SDRi (dB) |
| | | | Training | Inference | Training | Inference | |
| --- | --- | --- | --- | --- | --- | --- | --- |
| SepFormer | 50% | **26.0M** | 491 | 165 | 6.9 | 1.6 | 22.3 |
| **SepFormer (ours)** | 0% | 38.6M | **390** | **136** | **5.5** | **1.5** | **22.6** |

Since we do increase the number of Transformers by 50%, our model does have a significantly larger model size. Note, that the model size does not exactly increase by 50% since we only doubled the number of Transformers, but the rest of the architecture like the encoder and decoder stayed the same. The adjusted model is, however, also slightly more accurate, reaching 22.6 dB SI-SDRi compared to the original's 22.3 dB SI-SDRi.

Most importantly, however, are the differences in computation time and memory usage. During both training and inference, our model is roughly 20% faster. As for memory usage, during training there is again a roughly 20% reduction of memory usage while for inference both models perform very similarly. This decrease in computational cost is caused by our model only performing 75% of the original's sequence modelling steps. But because these sequence modelling steps are performed in sequence and therefore are aware of each other, it still reaches a higher separation accuracy.

The removal of overlaps can be thought of as a trade off between speed/memory usage and model size. We would, however, argue, that this trade off will basically always be worth it. In the case of the SepFormer model, the original model needs 100 MB while the adjusted needs 150 MB of disk space. We would argue, that an increase of 50 MB is irrelevant when compared to the speed and memory gains from the adjusted model.

The no overlap SepFormer model we show in Figure 4 and Table 1 is only one way to use our adjusted architecture. It is possible to match the number of sequence modelling steps by doubling the amount of Transformers to 64. This would, however, also cause a further increase in model size and would mean that the computational cost would be similar, if not slightly worse, compared to the original model. Memory usage would still be equal but the original model would be slightly faster due to performing two sequence modelling steps in parallel and our adjustment introducing some overhead like the additional positional encodings and decodings, sequence shifts and sequence dis- and reassemblies.

To summarize, there are various options for architectures that split sequences into chunks and operate on them in the time domain. If one wants to prioritize model size, high overlaps should be used. If accuracy, speed or memory usage are a priority, no overlaps should be used and the saved computational cost should instead be invested into making the model bigger by increasing the layer count or increasing the channel size..

### 3.3 AUDIO SUPER RESOLUTION - NU-WAVE2

The NU-Wave2 architecture is very different from the speech separation model covered in the previous section. Unlike the SepFormer, it only uses overlapped chunks for one part of its recurring block design while operating on the entire sequence for the rest of the model.

More specifically, the NU-Wave2 architecture uses the STFT and inverse STFT within each recurring block with an overlap ratio of 75%, meaning this step almost quadruples the size of the tensor. As we have previously stated, overlaps in the frequency domain cannot be treated the same as in the time domain as they serve a different purpose. One purpose is to minimize spectral leakage, while the other is to maximize spectral and temporal resolution of the STFT. Choosing a large overlap ratio means that a big window size is used which causes high spectral resolution and a low hop size is used which causes high temporal resolution. Removing overlaps from the STFT means reducing either the temporal or spectral resolution.

In order to reduce or even remove overlaps in the NU-Wave2 architecture, we cannot use all of the same tools as we used during the SepFormer adjustment. The main difference is that undoing and redoing the STFT is not a lossless step if overlaps are removed.

Another reason is that for the STFT, the NU-Wave2 block only applies sequence modelling along the frequency axis, not the time axis. Therefore we only have to match the size of the frequency axis while reducing the size of the time axis is irrelevant, meaning that reducing temporal resolution is not an issue in this particular case. If it were, the same solution that was used in the SepFormer adjusted architecture can be utilized where the STFT would have to be undone and redone for each sequence modelling layer and different window sizes for the spectral and temporal sequence modelling would be used in order to maximize their resolution. The only issue where this is not possible is if both axes would be processed simultaneously, like in the case of a two dimensional convolutional layer. In that case, one would have to choose between sacrificing spectral or temporal resolution or alternate between maximizing either the size of the frequency or time axis of the STFT for each sequence modelling layer.

While overlaps in the frequency domain are meant to minimize spectral leakage, in many model architectures, including the NU-Wave2, this is not a necessary precaution. There are a number of reasons for this, but the biggest one is that the STFT is part of a residual path, meaning that its output will be added to the input. While no distortion would still be preferable, some distortion in the residual path is not necessarily going to affect the output and can be corrected by the neural network. If the final output of the model were directly produced by the inverse STFT, then removing overlaps is ill-advised due to said distortion. The same is true for the window function. The original model uses a Hann window, while our adjusted model does not use a window function at all.

Another difference is that we cannot simply increase the number of blocks as we did in the Sep-Former because the recurring blocks contain other parts outside of the STFT, meaning there is no direct trade off between reducing overlaps and increasing the number of blocks. Furthermore, within the STFT context the NU-Wave2 is using one dimensional convolutional layers instead of Transformers for its sequence modelling. This means that positional encoding is not used here and therefore no major architecture changes are necessary.

The only architectural changes that were made is to once again shift the sequence before the STFT, set the hop size equal to the chunk size to remove overlaps, increase the hyperparameters of the layers operating on the STFT and finally undo the sequence shift after the inverse STFT. We set the shift value to start from 0 and increase by 10 for each STFT iteration. Since the window size of this model is much larger than the SepFormer, one might expect to need a larger shift value as well, but in our experiments it made no noticeable difference. As for the hyperparameters, we increased the kernel sizes of the one dimensional convolutional layer within the BSTFT layer from 3 to 5 and also increased the BSTFT channel size from 64 to 128. We use a custom STFT implementation since the builtin version in pytorch does not allow for zero overlap.

Table 2: Comparing the model size, log-spectral distance (LSD), computation time and memory usage on the VCTK corpus for models using overlap and models not using overlap. The LSD is reported for an input sampling rate of 8 kHz, 12 kHz, 16 kHz and 24 kHz.

| Method | Overlap | Model size | Computation time (s) | | Memory usage (GB) | | LSD | | | |
|---|---|---|---|---|---|---|---|---|---|---|
| | | | Training | Inference | Training | Inference | 8kHz | 12kHz | 16kHz | 24kHz |
| NU-Wave2 | 75% | **1.7M** | 6420 | 3770 | 42.1 | 8.8 | 1.14 | 1.01 | 0.925 | 0.774 |
| **Nu-Wave2 (ours)** | 0% | 2.6M | **3840** | **2900** | **33.8** | **7.1** | 1.15 | 1.02 | 0.93 | 0.796 |

Table 2 shows the comparison between the original NU-Wave2 and our adjusted version tested on the VCTK benchmark. Aside from the overlap ratio, model size, computation time and memory usage, we show the log-spectral distance (LSD) as it is the primary evaluation metric for audio super resolution. A lower LSD value represents a better score. The evaluation is shown for different upsampling scenarios, from 8, 12, 16 and 24 kHz to 48 kHz. The original paper includes some additional evaluation metrics, specifically the signal-to-noise ratio (SNR) as well as high-frequency and low-frequency LSD, which can be found in the appendix.

Once again, the model size of our adjusted version increased since an increase in hyperparameters is a core part of our strategy. However, since the original model already had very few trainable parameters, this increase is, in our opinion, irrelevant. This is especially true considering that the model itself is only takes up a few MBs of disk space but uses over 10 GBs of memory, meaning it is not an extremely lightweight model meant to run on low power devices.

Our adjusted model reaches slightly higher LSD values, meaning that the original model achieves better accuracy. This difference in accuracy, however, is only very minor and our adjusted model is significantly faster and more memory efficient instead. Specifically, computation time is lowered by 41% during training and 23% during inference while memory usage is lowered by 20% during training and 19% during inference.

## 4 CONCLUSION

In this paper, we challenged the common notion of using 50% or even 75% overlaps for time or frequency domain chunking. Overlapping chunks in the context of machine learning models are convenient since they fix issues that occur when splitting the sequence into chunks and will, using an identical model, outperform a model without overlaps. The cause of this is that each sequence modelling layer applies sequence modelling multiple times for each input since each input is contained multiple times due to overlaps. Overlaps are a form of parallelization, but they are not optimal because each of these sequence modelling applications has no awareness of the other sequence modelling applications within the same layer. The more effective way for model architectures is to instead remove or reduce overlaps and use the gained computational budget to increase the model size - that way each sequence modelling step will have awareness of each other sequence modelling step.

We describe two strategies for removing or reducing overlaps, including shifting the sequence and using different window sizes. Furthermore, we implemented our concepts on two different models, one in the time domain and one in the frequency domain and for two different problem areas.

We were able to significantly improve computation time and memory usage while maintaining accuracy for both of the models we tested. This does come at the cost of model size, however, we believe this to be a much less relevant metric than the other three.

For future work, we intend to test our methods of removing overlaps on more models. From our experiments and understanding of overlaps, it should be possible to remove them in nearly all architectures in the time domain. The frequency domain, however, needs to meet certain conditions to be able to remove overlaps completely. We would, however, still argue for at least reducing overlaps as currently most architectures using the STFT operate with an overlap ratio of 75% or even greater.

## 5 REPRODUCIBILITY STATEMENT

The general concepts for removing overlaps are described in section 2.

The specific implementations of our experiments are detailed in sections 3.2 and 3.3. The original code of the SepFormer (Subakan et al., 2021; 2023) can be found at https://github.com/speechbrain/speechbrain and the code for the NU-Wave2 (Han & Lee, 2022) model can be found at https://github.com/maum-ai/nuwave2.

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

## A   NU-WAVE2 - FULL RESULTS

Aside from the LSD values, we also include the SNR, LSD-LF and LSD-HF for comparison between the original model and our adjusted version in Table 3.

Table 3: Comparing the log-spectral distance (LSD), signal-to-noise ratio (SNR), low-frequency LSD (LSD-LF) and high-frequency LSD (LSD-HF) on the VCTK corpus for models using overlap and models not using overlap. The LSD is reported for an input sampling rate of 8 kHz, 12 kHz, 16 kHz and 24 kHz.

| Sampling rate | Metric | NU-Wave2 (original) | NU-Wave2 (ours) |
|---|---|---|---|
| 8 kHz | SNR ↑ | 18.8 | 18.7 |
| | LSD-LF ↓ | 0.219 | 0.25 |
| | LSD-HF ↓ | 1.24 | 1.25 |
| | LSD ↓ | 1.14 | 1.15 |
| 8 kHz | SNR ↑ | 21.6 | 21.5 |
| | LSD-LF ↓ | 0.275 | 0.33 |
| | LSD-HF ↓ | 1.15 | 1.15 |
| | LSD ↓ | 1.01 | 1.02 |
| 16 kHz | SNR ↑ | 24.0 | 24.0 |
| | LSD-LF ↓ | 0.305 | 0.376 |
| | LSD-HF ↓ | 1.10 | 1.10 |
| | LSD ↓ | 0.925 | 0.93 |
| 24 kHz | SNR ↑ | 28.4 | 28.6 |
| | LSD-LF ↓ | 0.326 | 0.42 |
| | LSD-HF ↓ | 1.02 | 1.01 |
| | LSD ↓ | 0.774 | 0.796 |

