# OpenReview forum: "On Sequence Segmentation with overlapped Chunks in Machine Learning"
_ICLR.cc/2025/Conference — Submitted to ICLR 2025_

### Official Review · Reviewer_z3L9 · 2024-11-01

**Soundness:** 1
**Presentation:** 1
**Contribution:** 2
**Rating:** 1
**Confidence:** 5

**Summary:**

This paper seeks to deepen the understanding of processing long sequences by means of independent processing of sub-sequences, which are taken as overlapping or non-overlapping chunks of the original sequence. After a short discussion concerning the use of chunked subsequences for sequence modeling, the paper takes the sepformer architecture as an example and demonstrates that by means of reducing the overlap processing time can be reduced. When considering a constant computational budget, the reduced computational costs may be invested into increasing the complexity of the model architecture, which finally leads to an improvement in the overall performance.

**Strengths:**

- The paper investigates the interesting question concerning the effect of overlap for transformer-based separation models, which treat the sequence of chunks independently.
- The most central claim of the paper: that a reduction of the overlap between subsequent chunks leads to a reduction in computational costs, and that in a situation of overall constant costs, this allows increasing the complexity of the model leaving to an overall improved performance has been experimentally demonstrated.

**Weaknesses:**

**Context**

The processing of chunked segments with overlap is a very old signal-processing strategy. For filtering and convolutional operations, this is called  Overlap-add processing. The approach has a long history, is conceptually simple, and theoretically very well understood. See https://en.wikipedia.org/wiki/Overlap-add_method with references in there (notably Oppenheim, Alan V.; Schafer, Ronald W. (1975). Digital signal processing. Englewood Cliffs, N.J.: Prentice-Hall). Looking into the discussion on Wikipedia one can note that there is the notion of border effects, which determine the theoretically optimal length of the overlap. The longer the filter response the longer the overlap has to be, if one wants to transparently replace the processing of the full sequence by means of the chunked processing. For autoregressive filters the response duration is infinite and a compromise between memory requirements and approximation has to be found. Note that the most efficient processing will remove all overlap. Unfortunately, this will introduce an incoherence at the edges, which in turn will be heard as annoying perceptual artifacts (clicks). That's why zero overlap is hardly ever used anywhere. In case one uses segment-wise processing with overlap-add the fundamental strategy to avoid border effects is cross-fading (see for example here Fröjd, Martin, and Andrew Horner. "Sound texture synthesis using an overlap–add/granular synthesis approach." Journal of the Audio Engineering Society 57.1/2 (2009): 29-37. On google you find numerous web pages for audio practitioners)

While  DNN (and transformers) are nonlinear the situation remains very similar. Two segments that are independently processed will be inconsistent at the edges. The deviation relates to the error the networks make compared to the optimal solution. These errors will, in most cases, be perceived as clicks. Clicks are one of the most annoying artifacts, and therefore overlap-add with cross-fades is perceptually nearly always better than not using overlap at all.  Before closing these fundamental comments, I'd like to note that there is an additional twist for DNN (including transformers). Given two separate blocks will lead to different results, and given the errors will often not be strongly correlated the cross-fade, or more generally averaging the different solutions, will lead to a slight advantage, similar to what is achieved when using ensemble algorithms (again see Wikipedia and references in there). For a practical example in the context of music source separation please see the shift trick in the first Demucs paper (https://arxiv.org/pdf/1911.13254). By means of extreme overlap, basically treating each segment 10 times, they achieve an SDR reduction of 0.3dB. Note this is not meant to say that this is the most efficient way to reduce the SDR. It is meant to underline that all these relations are very well known.
(I'd like to note that I am not involved in any of the references given.)

**Comments**

- The major part of the present paper, notably the introduction, discusses well-known facts (see the summary in the context section) that are widely used, not only in the signal-processing community but also in the machine-learning community.

- These well-known facts are presented in a way that contains numerous problematic descriptions and even errors:
  - down-sampling is a well-established procedure consisting of regularly dropping samples. The question related to what extent down-sampling is transparent and can be reversed is theoretically well understood (see https://en.wikipedia.org/wiki/Nyquist-Shannon_sampling_theorem and references in there).
  - Chunked processing without overlap is not down-sampling in the same way as described before. While the sample rate becomes lower (the frame rate in DSP terms), the sequence does not necessarily lose all context. The operation can be performed as a simple reshape which can be inverted. The question is how the model processes these segments. The context across the signal chunks is missing only, if the case where the segments are processed independently of each other. The context within the chunk is present in any case. When the DNN sees the sequence of chunks (as is the case for most models that work on STFT frames) then no context is missing, the context is just presented differently.  Note that even in case of overlap the context at the frame borders is still missing, but due to redundancy (each border is treated twice) and the uncorrelated errors (see shift effect above) this missing context can be partly compensated for.
  - While it is correct that wav2vec produces features with a reduced sample rate, wav2vec is not *down-sampling* either. The latent space that is produced by wav2vec does not consist of samples, it consists of features that do not aim to represent the full signal. Comparing these two procedures is highly misleading and would require clarification.
- In line 38 authors write: For other problem areas, mainly those which output not a class or text but audio itself, downsampling
that aggressively is not possible. These problem areas include audio generation, speech synthesis, source separation, speech enhancement and audio super resolution.

Processing of voice/music signals in the form of compressed representations (Encodec, DAC) is a very active research topic. While the representation has a much lower sample rate (so following the presentation in the current paper it qualifies as down-sampled), it does not necessarily have a negative impact and there is no problem of missing context. This is because the latent codes are not processed independently but as a sequence. The current research is a direct contradiction to the statement above.

- The presentation of the STFT is wrong.

*Line 158: The Fourier transform assumes that the signal is periodic. *

It is the Fourier Series that assumes the signal is periodic. The Fourier transform assumes the signal square integrable (https://en.wikipedia.org/wiki/Fourier_transform)

The STFT is not used because the signals to be analyzed are not periodic, but because most interesting signals (voice) are produced by time-varying systems, to observe the evolution of the frequency characteristics over time one has to cut into small pieces.

*line 160 : To minimize spectral leakage, the signal is multiplied with a window function whose
purpose it is to taper off the start and end of the signal to 0 making the signal somewhat periodic.*

No this is not to make the signal somewhat periodic. If you apply a time-limited window function the result remains non-periodic. Note as well that spectral leakage is introduced due to the cutting into frames. Cutting out signal frames without tapering is equivalent to using a rectangular window, which creates the most dramatic leakage. The tapering changes the way the spectral leakage will take place. For details on how various analysis windows modify spectral leakage see https://en.wikipedia.org/wiki/Window_function)

- The evaluation is insufficient to evaluate the results. As I have tried to explain in the introducing remarks for zero overlap it is likely that the inconsistencies at the segment borders lead to perceptually annoying artifacts (clicks). It may very well be that the perceptual comparison between 50% overlap and 0% overlap, together with a constant computational budget would result in the 0% variant coming out better (From my experience with this kind of algorithm, I would be astonished though). In any case, simply comparing objective measures is not sufficient to draw a conclusion.

**Questions:**

The article treats a valid question but does not use appropriate means to create any new insights. The introduction would need a major rewrite to clearly and correctly expose the research question differentiating between downsampling and segment-based processing, latent codes, and related models, and the special case of independent handling of segments or chunks. This section should include a correct discussion of the benefits  and disadvantages of the overlap between chunks ((see in the section context for ideas).

Given, the fundamental relations between reducing overlap and the potential to increase system complexity and therefore improve SDR are rather well known, but on the other hand, it is also known that reducing overlap will introduce specific artifacts (see the section context above), which are short (only at the edges) and therefore are not well reflected in the objective metrics, the experimental evaluation should include a second evaluation that validates the results for the intended final application. Here you should probably include different degrees of overlap (for example 50%, 25%, 10%, 5%, 0%) making sure that the computational costs increase such that overall all variants have the same compute budget. Concerning the computational costs you would best measure this in GFLOPS to not depend too much on specificities of a particular machine with a given memory.

In case the final application is an ASR algorithm the question would be whether the ASR algorithm works better with a slightly increased SDR, or with the inconsistencies (clicks) that arise at the chunk borders. This can be evaluated with word error rate for example. Note that lower SDR does not necessarily lead to lower word error rate.

In case the final application are human listeners, there should be a perceptual evaluation. Here you should ask the participants, how they perceive the quality of each of the different versions taking into account separation quality and artifacts. The Mean Opinion Score (MOS) could be used. Based on such a perceptual evaluation you should be able to conclude whether 0% overlap is the best variant for human listeners.

This discussion should make it clear that the SDR is an indicator but not a strong metric for final optimization of a separation algorithm.  What is best depends on what you want to do with the results. My personal guess is that for ASR no overlap may be the better choice, because the very local inconsistencies are not so dramatic for the word error rate, on the other hand regular clicks may stand out of all the other artifacts , and therefore be much more annoying for human listeners than the 0.3dB you loose if you use 50% overlap. I have made such experiments myself. It turned out that border effects remain rather limited for convolutional separation algorithms, with only 10% overlap and 1 second chunks I could significantly reduce the computational costs without any perceptual effects at the chunk borders.

---

### Official Review · Reviewer_1uQf · 2024-11-02

**Soundness:** 2
**Presentation:** 1
**Contribution:** 1
**Rating:** 1
**Confidence:** 5

**Summary:**

The paper discusses the inefficiencies of handling long sequences in machine learning models, particularly in audio tasks such as source separation or super-resolution. In these tasks, splitting sequences into overlapped chunks is commonly used to preserve contextual information. This overlapped processing method increases tensor size, redundant computations, and memory usage.

To address this and remove/reduce the overlap, the authors propose two strategies: random sequence shifting and variable chunk sizes between the network layers. These strategies ensure that context is not lost while being faster and less memory-hungry. However, this potentially leads to reduced performance due to inconsistencies in sequences between the layers, which the authors propose to overcome by increasing the model parameters, thus maintaining the model performance.

Experiments are conducted on two task-model combinations: 1) Speech Separation & SepFormer, and 2) Audio Super-Resolution and NU-Wave2. Experiments show reduced training/inference time and memory usage while maintaining model performance, albeit increasing model size and storage requirements.

**Strengths:**

-	New approach to audio sequence processing task.
-	Applies specific modifications to two model architectures and shows memory and time improvements, enabling larger models to be created to maintain or improve performance

**Weaknesses:**

**Limited Scope in Model Testing**: Although the paper tests two models, the scope remains narrow, limited to specific architectures for specific audio tasks. Moreover, the proposed methods require very careful engineering depending on the particular model. The dual-path model structure is a very specific architecture class used in source separation, there are several other models which do not apply overlapped sequence modeling, or even if they do, for example models working in spectrogram domain, as per the authors own discussions, it is not trivial to apply these methods directly to the STFT since it will break the invertibility.

**Trade-offs in Model Complexity**: By increasing parameters and layers to compensate for the lack of overlap, the model complexity and size are affected. While these modifications lead to computational efficiency, they come at the cost of increased model parameters, which may not always be viable for applications with strict size limitations.

**Insufficient Discussion on Frequency Domain Overlaps**: The approach in the frequency domain (NU-Wave2) is less straightforward and generalizable due to the complexities of handling spectral information. The proposed modifications reduce overlaps but also risk spectral leakage, making the findings less conclusive for broader applications.

**Limited Experimentation**:  They paper presents experimnetal results for each method-model on a single dataset and the results are without any confidence intervals. There is no attempt at evaluating the generalizbility of the proposed methods. There is no ablation study, e.g., how much the performance degrades with same model size, how does the performance degrade as the overlap is reduced, etc.

**Clarity**: The text descriptions are very dense and difficult to follow, since for each model the method is completely different, makes it even harder. While a lot of space has been given to overlapped waveform and STFT illustrations, the actual architecture diagrams are not very informative. Also, the more complex modifications of the NU-Wave2 architectural changes are not illustrated with a diagram.

**Lack of Training Details**: The paper lacks key details about the training infrastructure and hyperparameters for reproducibility.

**Questions:**

-	On what device were the computation times measured?
-	Does the training computation time include the backward pass as well or only the forward pass of the model?
-	For inference war overlap add still used for longer sequences?
-	What would be the effect of applying it to models which work on STFT domain, but do not do overlapped processing in the model itself?

---

### Official Review · Reviewer_okE4 · 2024-11-02

**Soundness:** 2
**Presentation:** 1
**Contribution:** 2
**Rating:** 3
**Confidence:** 3

**Summary:**

This paper proposes a group of techniques to avoid overlap when processing long audio sequences in chunks, including shifting the sequence, using variable chunk sizes and increasing model parameters, thereby reducing computational complexity. The author conduct experiments on speech seperation and audio super-resolution to validate the effectiveness of their approach.

**Strengths:**

Experiments on speech seperation and audio super-resolution proves that the proposed tricks do reduce computational complexity in long audio processing.

**Weaknesses:**

1. Regarding the increase in model size leading to performance improvements as a significant contribution seems to be not persuative, since it's more like a common sence that a larger parameter count enables the model to capture more complex probability distributions, thereby enhancing its performance.

2. Did the authors investigate previous works related to reducing computational overhead in processing long audio sequences? This aspect is not reflected in the text. I suggest adding an extra chapter of related works, and describing how do the proposed methods compare to earlier work in terms of advantages or differences

3. No enough ablation studies are provided to validate the effectiveness of the proposed three methods. Does each of them contribute to performance improvements, or is it solely the increase in model parameter count that proves beneficial? I suggest that the authors add experiments that introduce shifting the sequence and using variable chunk sizes separately, while keeping the model architecture and parameter count unchanged. They may also increase the model parameter count without incorporating these methods and report the results to validate the effectiveness of each technique.

4. The paper seems to be not well organized, and the narrative logic is somehow disorganized.

   4.1. The first chapter extensively discusses overlap instead of providing an introduction or overview of the whole paper. I suggest the authors to use the first chapter to briefly outline the background of the problem, the proposed methods, and their contributions (i.e., an overview of the entire paper). Following this, a separate chapter could be added to analyze the overlap issue in segmenting long audio sequences.

   4.2. In section three, the authors use lengthy descriptions on the modifications to the speech separation model, while the architecture illustrations are overly brief, providing only a sequence of modules without detailing each one, which makes it difficult for readers to understand the design. The authors could consider replacing lengthy descriptive text with more detailed architectural diagrams to enhance clarity.

**Questions:**

Did the authors attempt subjective evaluations of the generated results? Are there perceived differences, and are there samples available?

---

### Official Review · Reviewer_iZYR · 2024-11-04

**Soundness:** 3
**Presentation:** 2
**Contribution:** 3
**Rating:** 5
**Confidence:** 3

**Summary:**

Sequence modeling techniques such as Transformers scale quadratically with input sequence lengths so it is not viable to apply transformers to long speech sequences. The long inputs are split into equal-sized chunks and then sequence modeling is applied to much shorter chunks. To avoid losing the context that exists between chunks, the sequence is divided into overlapping chunks instead. However, this duplicates the information that exists between chunks and increases the computational cost.

The paper proposes an alternative to overlapped chunks for processing long speech sequences. The paper describes two strategies for removing or reducing overlaps:  shifting the sequence and using different window sizes. By removing the overlap between the chunks, the models effectively process less data and this significantly improves computation time and reduces memory usage.

**Strengths:**

The paper is unique and interesting. It explores alternatives to commonly accepted overlapped chunking for long speech sequences. The proposed techniques achieve a significant reduction in runtime and memory usage for two different tasks: speech separation and audio super-resolution.

**Weaknesses:**

The writing and the organization of the paper can be improved. The paper feels bloated in a few places e.g. there is no need for Figure 1,2 to be almost full pages. There is no need for five examples of chunking. At the same time, some experimental details are missing. For example, "We increase the shifts by 10 for each Transformer starting from 0. In our experiments changing the shift value had only minor impact on accuracy as long as it was not too small in reference to the chunk size."

What is too small? At what point does the shift value not impact the final performance? What is the final chunk size used in the experiments?
Including a graph can help answer these questions. The experimental section would benefit from ablation experiments concerning different adjustments made to the modified models.

**Questions:**

1) L345: "The third adjustment is to choose different chunk sizes for the intra- and inter-processing." What is the performance with the same chunk sizes?

2) L367: "The fourth and final adjustment to the SepFormer architecture is to simply increase the total number of Transformers." What is the performance with the same number of parameters?

3) What is the impact of chunk size on the model performance?

4) L334: "In order to fix this issue, we simply subtract the positional embeddings at the end of each Transformer." Transformers are non-linear functions, how does simply subtracting positional embeddings from the transformer's outputs perform positional decoding?

5) For the reasoning for choosing the same chunk sizes as the original sepformer L364 argues that: "While this would be the more practical solution, it also makes comparisons with the original model less meaningful since any change could be caused by the difference in the sizes of the sequence axes" Could the reader not make a similar argument that comparisons with the original model are less meaningful because of other factors such as parameter counts, the way positional embeddings are applied?

---

### Meta-Review · Area_Chair_d6dv · 2024-12-16

**Metareview:**

This paper addresses the problem of long-sequence speech modeling. Traditionally, long speech sequences are divided into smaller overlapped chunks, which increases computational cost. This paper explores reducing or eliminating overlap to address this issue. Any resulting performance degradation is mitigated by increasing model parameters.
However, there are several issues with this submission.
The authors missed the fundamental context of overlap processing of speech signals which comes from a very old signal processing strategy: overlap add. Reviewer z3L9 provided very detailed context of why the overlap between chunks is necessary. Without overlap, there will be discontinuities across chunks, introducing short-period perceptual artifacts.
Increasing the model size to mitigate the performance drop by the proposed methods is not convincing as it is common sense that the model performance usually improves with more parameters. The performance drop with the same number of parameters indicates that the proposed method does not fundamentally solve the problem.
Simply using SDR as the metrics may not be the right way to evaluate the proposed methods as the short-period perceptual artifacts may not affect the final metrics too much. Depending on the downstream application, the authors should evaluate WER for ASR tasks, or introduce subjective evaluation for human listening.

**Additional Comments On Reviewer Discussion:**

There is no author/reviewer discussion.

---

### Decision · Program_Chairs · 2025-01-22

Reject